# Genomic Dissection of a Wild Region in a Superior *Solanum pennellii* Introgression Sub-Line with High Ascorbic Acid Accumulation in Tomato Fruit

**DOI:** 10.3390/genes11080847

**Published:** 2020-07-24

**Authors:** Antonietta Aliberti, Fabrizio Olivieri, Salvatore Graci, Maria Manuela Rigano, Amalia Barone, Valentino Ruggieri

**Affiliations:** 1Department of Agricultural Sciences, University of Naples Federico II, 80055 Portici (Naples), Italy; antonietta.aliberti@unina.it (A.A.); fabrizio.olivieri@unina.it (F.O.); salgra95@gmail.com (S.G.); mrigano@unina.it (M.M.R.); ambarone@unina.it (A.B.); 2John Innes Centre, Norwich Research Park, Norwich NR4 7UH, UK; 3Biomeets Consulting, ITNIG—Carrer d’Àlaba, 61, 08005 Barcelona, Spain

**Keywords:** genome reconstruction, genotyping-by-sequencing, RNA-Seq, introgression lines, red ripe fruit, *Solanum lycopersicum*

## Abstract

The *Solanum pennellii* introgression lines (ILs) have been exploited to map quantitative trait loci (QTLs) and identify favorable alleles that could improve fruit quality traits in tomato varieties. Over the past few years, ILs exhibiting increased content of ascorbic acid in the fruit have been selected, among which the sub-line R182. The aims of this work were to identify the genes of the wild donor *S. pennellii* harbored by the sub-line and to detect genes controlling ascorbic acid accumulation by using genomics tools. A Genotyping-By-Sequencing (GBS) approach confirmed that no wild introgressions were present in the sub-line besides one region on chromosome 7. By using a dense single nucleotide polymorphism (SNP) map obtained by RNA sequencing (RNA-Seq), the wild region of the sub-line was finely identified; thus, defining 39 wild genes that replaced 33 genes of the ILs genetic background (cv. M82). The differentially expressed genes mapping in the region and the variants detected among the cultivated and the wild alleles evidenced the potential role of the novel genes present in the wild region. Interestingly, one upregulated gene, annotated as a major facilitator superfamily protein, showed a novel structure in R182, with respect to the parental lines. These genes will be further investigated using gene editing strategies.

## 1. Introduction

The cultivated tomato (*Solanum lycopersicum*) belongs to one of the most important and large plant families worldwide. Indeed, the Solanaceae family consists of about 98 genera and 2700 species [1]. In addition to the cultivated species, there are other related wild species, including *Solanum pimpinellifolium*, *Solanum chmielewskii*, *Solanum chilense*, *Solanum neorickii*, *Solanum peruvianum*, *Solanum habrochaites*, and *Solanum pennellii* [2,3]. Recent studies have focused on S. *pennellii,* a wild species from South America, considered an important donor of germplasm for the cultivated tomato, which lost genetic variability during domestication and evolution [4]. In order to fully exploit the genetic variability of wild species, introgression lines (ILs) were obtained, which carry homozygous regions of the wild genomes in a common cultivated background. An entire set of ILs represent a wild species’ entire genome and can be used to transfer traits into commercial cultivars. Up until now, the most widely exploited IL population for genetic and molecular studies was that obtained by crossing *S. pennellii* with the cultivated variety M82, consisting of 76 introgression lines and sub-lines [5,6,7,8]. These lines are an excellent source of genetic variation, so far used for the identification of more than 2700 agronomical useful Quantitative Trait Loci (QTLs), such as those controlling plant biomass yield, drought tolerance, chemical composition of the fruit, and commercial traits of interests [9]. In particular, in recent years, researchers have focused their attention on QTLs related to the nutritional quality of tomato fruits, and to their phytochemicals and antioxidant compounds, including carotenoids (mainly lycopene and β-carotene), phenolic compounds, and vitamins C and E [10,11]. Among these compounds, vitamin C (or ascorbic acid, AsA), shows a powerful antioxidant activity in the human body, and has been implicated in inflammation prevention and DNA protection from damages induced by antioxidant species’ [12]. For most of the metabolites present in tomato fruits, including carotenoids and some phenolics compounds, the genetic and molecular mechanisms leading to their biosynthesis and accumulation in tomato fruit are already known [13,14]. As for AsA, the biosynthetic pathways controlling its synthesis and accumulation are well-characterized [15], but not all upstream regulators in tomatoes are known.

Recently, the use of next-generation sequencing (NGS) led to a better comprehension of the genetic basis of many biological processes in plant life [16,17], and allowed the identification and fine detection of QTLs and candidate genes controlling these processes [18,19]. Among these methods, Genotyping-By-Sequencing (GBS) is one of the most powerful and affordable strategies for genome-wide genotyping and single nucleotide polymorphisms (SNPs) discovery. The approach relies on the use of restriction enzyme digestion to reduce genome complexity before sequencing [20,21,22]. In addition, RNA sequencing (RNA-Seq) is commonly used to analyze gene expression and to uncover novel RNA species [23], and has allowed identifying candidate genes controlling different traits [24,25]. RNA-Seq analysis has already been used in tomatoes to identify differentially expressed genes (DEGs) involved in AsA, carotenoid, and flavonoid biosynthetic pathways [26,27]. Although RNA-Seq is primarily considered a method for gene expression analysis, it can also be used to identify genomic variants in expressed regions alongside whole-exome (WES) and whole-genome sequencing (WGS) [28].

In recent years, in our laboratory, several studies focused on a group of *S. pennellii* ILs selected for their high AsA content in the fruits and investigated using different genomics tools [4,6,29,30,31]. Among these ILs, the line IL7-3 was selected, which carries an introgressed region on chromosome 7. Additionally, a group of sub-lines was obtained, which carried reduced portions of the wild genome with respect to IL7-3 on the same chromosome [11,32,33,34]. In particular, the sub-line R182 was selected for its better performances in terms of fruit quality. Indeed, compared to the parental lines, the sub-line combined the high AsA level derived from IL7-3 with good yield performances derived from M82, also exhibiting a very high firmness of the fruit, significantly superior to that of both parents. Indeed, this sub-line showed an average increase of more than 30% of AsA, 25% of °Brix, and 22% of firmness, respectively, compared to the cultivated line M82, whereas it always exhibited a yield level comparable to that of M82, thus far exceeding the low yield always found in the parental line IL7-3 [35].

The introgressed fragment size of the sub-line R182 was previously defined by Sequence Characterized Amplified Region (**SCAR**) and Cleaved Amplified Polymorphic Sequence. (**CAPS**) molecular markers; however, the role of some candidate genes (CG) mapping in the introgressed region needed to be further defined [32,35]. In order to better investigate the involvement of wild alleles introgressed in the sub-line R182 in increasing AsA content, the aim of the present work is (1) to verify the purity of R182 line, excluding the presence of spurious wild fragments spread all over the genome; (2) to correctly define the introgressed region and the genes of the wild donor *S. pennellii* harbored by the sub-line; and (3) to transcriptionally characterize the candidate genes controlling AsA accumulation. By exploiting high-throughput genomics tools, such as GBS and RNA-Seq, we reconstructed the wild genome of the introgression sub-line R182, and clearly identified the wild alleles mapping in the introgressed region and putatively involved in controlling AsA accumulation in R182 red ripe fruit.

## 2. Materials and Methods

### 2.1. Plant Materials

Plant material consisted of the *S*. *pennellii* introgression line IL7-3 (accession LA4066), the IL7-3 sub-line R182, and the cultivated genotype M82 (accession LA3475). The sub-line R182, kindly provided by Dr. Dani Zamir (Hebrew University, Israel), was selected by using species-specific markers at the Department of Agricultural Sciences of University of Naples Federico II, and then characterized considering different fruit quality traits [32,35]. The three genotypes were grown in open field during the year 2017 in a randomized scheme, using three replicates per genotype, and 10 plants per each replicate, and following the traditional farming practices in the geographical area. A pool of 10 fruits from the genotypes M82 and R182 was collected from different plants at two development stages: breaker (BR, 45 days post anthesis) and mature red (MR, 55 days post anthesis). For each sample, seeds and columella were removed and fruits were ground in liquid nitrogen and stored at −80 °C until analyses. Leaves were collected from seedlings of IL7-3, M82, and R182, ground in liquid nitrogen by mortar and pestle to a fine powder, and stored at −80 °C until DNA genotyping analyses.

### 2.2. GBS Analysis

Genomic DNA was extracted from 100 mg of young leaf tissue from R182 and the parental lines M82 and IL7-3 using the DNeasy plant mini kit (Qiagen). DNA concentration was determined by using a fluorometer (Invitrogen, Carlsbad, CA) while the 260/280 and 260/230 ratios were measured by using a NanoDrop spectrophotometer (Thermo Fischer Scientific, Waltham, MA, USA). For DNA sequencing, 1 µg of DNA diluted in 30 µL of sterile Milli-Q water was used to obtain the libraries for the ddRAD technology, as reported by Peterson et al. [36]. The restriction enzymes *Mbo*I and *Sph*I were used for DNA digestion and the fragments were sequenced using the V4 chemistry paired end 125 bp mode on a HiSeq2500 instrument (Illumina, San Diego, CA, USA). The pipeline Stacks v2.0 [37] was used to demultiplex, clean raw Illumina reads and detect the variants. The reads alignment to the reference genome of *Solanum lycopersicum* (Tomato Genome version SL3.0) was performed using BWA-MEM [38] with default parameters. Raw variants were filtered using VCFtools v.0.1.13 (http://vcftools.sourceforge.net) [39]. The minimum mean of Depth of Coverage (min-mean DP) = 5 was used as filtering parameter. Gene annotation and prediction of the possible effect of the SNP mutations were evaluated by the SnpEff tool (http://snpeff.sourceforge.net/) [40], using the tomato genome assembly SL3.0 and the iTAG3.2 annotation as references.

### 2.3. RNA-Seq Analysis

RNA extraction was performed from M82 and R182 fruits collected at breaker and mature red developmental stages by using the TRIzol reagent (Thermo Fischer Scientific, Waltham, MA, USA) and following the manufacturer’s guidelines with some modifications. One mL of TRIzol reagent was used for 100 mg of frozen tomato fruit powder, afterwards the mixture was vortexed, incubated on ice for 5 min, and centrifuged at 14,000 rpm for 10 min at 4 °C. The supernatant was transferred in a fresh Eppendorf tube and stored on ice for 5 min. Then, 0.2 mL of chloroform were added, the mixture was vortexed, incubated for 5 min on ice, and centrifuged at 14,000 rpm at 4 °C for 15 min. The colorless upper aqueous phase was transferred in a fresh Eppendorf tube, 0.5 mL of 2-propanol was added, the mixture was gently mixed by pipetting, incubated on ice for 10 min and centrifuged at 14,000 rpm for 10 min at 4 °C. The supernatant was discarded; the RNA pellet was washed four times by adding 1 mL of 75% ethanol, vortexed, and centrifuged at 14,000 rpm for 5 min at 4 °C. The pellet was then briefly dried for 5–10 min and dissolved in DEPC-treated water. Then, 0.1 V of 3 M Sodium Acetate (NaOAc, pH 5.5) was added and the RNA solution was gently mixed by pipetting. After that, 2 V of 100% ethanol were added, and the solution was gently mixed. For RNA sequencing, 3 µg of RNA diluted in 50 µL of DEPC-treated water were used and the experiment was performed by the “TruSeq mRNA” protocol for preparing libraries, followed by a pair-end strategy for sequencing. The differential expression analysis was carried out using the Sequentia Biotech data analysis software “AIR” (https://transcriptomics.sequentiabiotech.com/), which relies on the Bioconductor package edgeR [41]. Genes were considered significantly differentially expressed if the false discovery rate (FDR) of the statistical test was less than 0.05.

### 2.4. R182 Introgression Analysis and Genome Reconstruction

The RNA-Seq reads obtained from the genotype R182 were used to identify variants against the two parental genomes (*S. lycopersicum* M82 and *S. pennellii*). In particular, the variants obtained against the M82 genome (the background of the sub-IL) allowed identifying the break points of the introgressed region, whereas the variants identified against the *S*. *pennellii* genome (the wild donor of the introgression) allowed defining the size of the introgressed region. In order to map the variants, refine the mapping on the splicing junctions, and call the variants on the two genomes, three software were exploited using default parameters: STAR [42], OPOSSUM [28], and PLATYPUS [43]. Then, heterozygous and low-quality variants were filtered out using VCFtools [39]. The genome of the R182 was reconstructed by merging the M82 genome, defined by the break points with the *S. pennellii* complementary segment. The corresponding gene features coordinates of both parental lines were transferred to the reconstructed R182 genome by a lift-over process.

For comparative genomic analyses, the Best Bi-directional Hits (BBH) between genes of M82 and *S. pennellii* were assessed by InParanoid v8 [44] software using as input the complete proteins set of *S. lycopersicum* iTAG4.1 and the *S. pennellii* v2 annotations. For the genes that did not show a BBH, a BLASTp [45] search was conducted (e-value 1 × 10^−10^) to evaluate the best match. A synteny analysis between the parental lines was also performed by using SyMAP v.5 [46].

### 2.5. Molecular Marker Analysis

A border check reliability was carried out by looking at the expression of the genes in RNA-Seq data around the putative borders of the introgression. When the genes were not expressed (no reads were available for identifying polymorphisms) specific SCAR/CAPS markers were designed in order to better identify the genes of the wild region introgressed into the R182 genome. Amplification of genomic DNA was carried out using primers designed based on polymorphisms detected in the R182 introgressed region between *S. lycopersicum* (iTAG4.1) and *S. pennellii* (v2) genomes. PCR amplification was carried out in 50 µL reaction volume containing 50 ng DNA, 1X of MyTaq^TM^ reaction buffer, 1.0 mM primer, and 1 U MyTaq^TM^ DNA polymerase (Bioline). For designing CAPS markers, restriction enzymes suitable to detect polymorphic SNPs between the fragments amplified were found using the tool CAPS Designer available at the Sol Genomics Network (solgenomics.net). Amplified and restricted fragments were visualized on agarose gel at different concentrations depending on their expected size. In order to identify if the last gene of the introgressed region is present in its wild (Sopen07g024640) or cultivated (Solyc07g049310) allele, the whole gene was sequenced in the genotype R182. The amplification was carried out using three primers pairs (Appendix A) designed based on the Sopen07g024640 genomic sequence. PCR amplification was carried out in 50 µL reaction volume containing 50 ng DNA, 1X of Phusion HF buffer, 1.0 mM primer, 10 mM dNTPs and 1 U Phusion™ High-Fidelity DNA Polymerase (Thermo Scientific™). PCR products were analyzed on 1% (w/v) agarose gel and amplicons obtained were purified using the QIAquick Gel Extraction Kit (Qiagen). Then PCR purified products were sequenced using the Eurofins sequencing service (Mix2Seq kit). Geneious software v11.1.5 (Biomatters, http://www.geneious.com) was used to process sequences and MUSCLE algorithm to perform the multiple sequence alignment (https://www.ebi.ac.uk/Tools/msa/muscle/).

## 3. Results

### 3.1. GBS Analysis

In order to better characterize the sub-line R182, the presence of spurious (outside of the introgressed region) wild fragments in its genome was ascertained by analyzing data coming from a wider GBS experiment, including the sub-line R182, IL7-3 and the cultivated parental genotype cv. M82. As total, 458 filtered polymorphisms (Appendix A) were extracted. Among these, just six (1.3%) mapped outside of the introgressed region 7–3, one on chromosome 1, one on chromosome 4, and two on chromosomes 5 and 12, respectively. The remaining 452 variants, as expected, mapped within the wild region 7–3, being 365 (80.8%) SNPs and 87 (19.2%) insertions or deletions (INDELs). R182 genotyping showed 14 SNPs, and four INDELs in a well-defined region of chromosome 7 extending from SNP at position 59,347,691 (Solyc07g048020) to SNP at position 59,641,222 (Solyc07g049230). The result confirms that no spurious wild introgressed regions were present in the parental line IL7-3 neither in its sub-line R182. The SnpEff analysis of the 18 mutations of R182 evidenced that they affected the genes Solyc07g048020 (two SNPs and one INDEL), Solyc07g048060 (seven SNPs), Solyc07g049160 (one SNP and two INDELs) and Solyc07g049230 (four SNPs and one INDEL). In most cases, these mutations caused modifier effects on the proteins, being classified as intergenic variants.

### 3.2. RNA-Seq Analysis of R182 and DEGs Identification

A RNA-Seq analysis was carried out to highlight transcriptomic differences between the sub-line R182 and the parental genotype M82. Two stages of fruit ripening were assessed, the breaker (BR) and mature red (MR). The quality check (QC), performed on the raw sequencing data to remove low quality portions and Illumina adapters, showed an average reduction of 15% of reads number (Appendix A). The remaining high-quality reads aligned against the *S. lycopersicum* reference genome showed an average of uniquely mapped reads of approximately 94% and an average mismatch rate *per* base of 0.14% (Appendix A).

Following the RNA-Seq data analysis, the R HTSFilter package was applied for the statistical analysis aimed at removing the not expressed genes and the ones showing a high variability. The software relies on a filtering procedure for replicated transcriptome sequencing data based on a Jaccard similarity index. As a result, 19,515 and 16,444 genes were retained for BR and MR samples, respectively. The genes passing the HTSFilter were then used for differential expression analysis (Figure 1A). The comparison between R182 and M82 for the BR stage showed 54 differentially expressed genes (DEGs) of which 35 upregulated and 19 downregulated, whereas for the MR stage evidenced 100 DEGs of which 40 upregulated and 60 downregulated (for a complete list of the DEGs see Appendix A). As a whole, the DEGs identified in BR and MR fruit mapped across all the chromosomes (Table 1), with a higher number on chromosomes 1, 3, and 7. The most significant differences in expression at BR stage (Figure 1A) were highlighted for Solyc03g045140 (Cyclopropane-fatty-acyl-phospholipid synthase), Solyc09g007020 (Pathogenesis-related protein) as well as for Solyc07g048100 (BRCT domain-containing protein) and Solyc07g062600 (Acyl-CoA N-acyltransferase). At MR stage (Figure 1B) the most significant expression change was recorded for Solyc07g049140 (Metallocarboxypeptidase inhibitor), which appears to be highly downregulated (-13 log2FC) in R182 compared to M82.

Searching among genes commonly regulated in both the studied fruit developmental stages (BR and MR), seven upregulated and eight downregulated genes were found, five of which mapped in the introgressed region of the sub-line R182 (Table 2). Another gene (Solyc07g049290) of the introgressed region of R182 resulted upregulated, even though only in the MR stage; and it is annotated as a major facilitator superfamily (MFS) protein-like. The DEGs were also functionally annotated to analyze their gene ontologies (GO). DE genes at BR stage highlighted GOs enriched for plant cell wall organization, proteolysis and for fatty acid metabolism terms. By contrast, GO enrichment for MR stage showed, among others, an abundance of terms related to amino acid and sucrose metabolic processes (Appendix A).

### 3.3. R182 Introgressed Genome Reconstruction

In order to define the group of wild genes mapping in R182 introgressed region, we took advantage of the expressed reads obtained from the RNA-Seq analysis, by identifying those that were polymorphic between R182 and M82. In particular, in this analysis a gene was considered wild (from *S. pennellii)* in R182 if the number of variants that it harbors is greater than the number of variants found in the cultivated genome background (M82). A check across the whole genome (Appendix A) clearly showed as only one single region on chromosome 7 presents wild genes in the R182 sub-line confirming data reported by the GBS analysis. Specifically, an interval including a group of 31 genes, from Solyc07g048010 to Solyc07g049310, showed an average of 17 variants *per* gene indicating that they come from the wild donor (Appendix A). The same analysis was carried out by comparing the reads of R182 across the *S. pennellii* genome. In this case, to confirm that a wild gene was introgressed in the R182 sub-line we do not expect to detect variants respect to the wild genome. The result confirmed the presence of two segments with wild genes in the R182 genome, one including a group of genes spanning from Sopen07g024420 to Sopen07g024640 and the second including another group of 16 genes from Sopen07g025130 to Sopen07g025280 (Appendix A).

Figure 2 shows as these two splitted wild regions are syntenic with one region identified on M82. However, in order to clarify this unexpected result and to exclude mis-assembly issues of the *S. pennellii* genome, we analyzed more in details the homology of the genes mapping in the region. First of all, we evaluated the correspondence on the basis of their protein and genomic alignments; thus, detecting the homologous relationships between *S. pennellii* and *S. lycopersicum* genes belonging to the introgressed region of R182 sub-line (Appendix A). Twenty-two Best Bidirectional Hits (BBH) relationships were found pointing out the high similarity/orthology between the two parental lines in this region (Table 3). However, some differences were also observed. In particular, three *S. lycopersicum* (Solyc07g048020, Solyc07g049135 and Solyc07g049215) and seven *S. pennellii* (Sopen07g024570, Sopen07g024580, Sopen07g025150, Sopen07g025180, Sopen07g025190, Sopen07g025200, Sopen07g025250) genes appeared to be species-specific since they did not show any homology compared to annotated genes of the contrasting genome. In addition, from this analysis we could also report a one-to-many relationship between a *S. pennellii* gene (Sopen07g024560) and two *S. lycopersicum* genes (Solyc07g049120, Solyc07g049130) underlying a probable duplication event in *S. lycopersicum* cv. M82. Probable duplication events were observed in the opposite direction as well, as for example between one *S. lycopersicum* (Solyc07g049240) and two *S. pennellii* (Sopen07g025260, Sopen07g025270) genes. After, we combined this analysis with the information obtained from eight species-specific markers built to confirm missing points in the polymorphism analysis obtained by RNA-Seq data (due to absence of expression/available reads). Altogether, these data allowed reconstructing the presence and order of the wild genes of the introgressed region of the sub-line R182 (Figure 2). Comprehensively, in the R182 sub-line 39 wild genes replaced 33 *S. lycopersicum* cv. M82 genes. Marker analysis confirmed that no other region between the wild segments was introgressed, thus also confirming the synteny of the wild segments with the corresponding *S. lycopersicum* segment.

### 3.4. Sopen07g024640 Sequencing

Since the molecular marker analysis carried out on the last gene of the introgressed region (Solyc07g049310/Sopen07g024640) did not allow to clearly identify the presence of the cultivated or wild allele, we carried out the sequencing of Sopen07g024640 (8928 bp) of the R182 genotype, dividing this gene in three parts spanning from 70,111,740 to 70,114,744 bp, from 70,114,745 to 70,117,772 bp and from 70,117,773 to 70,120,667 bp. Results from the Sanger sequencing showed that the first two parts of the Sopen07g024640 sequenced in R182 perfectly align with the sequences of *S. pennellii*. As for the third part, the multi-alignment between the Sopen07g024640, the ortholog Solyc07g049310 and the corresponding sequence in R182, showed at position 70,119,958 the last *S. pennellii* variant of the R182 sequencing product, whereas the first *S. lycopersicum* variant was detected at position 70,120,258 bp (Figure 3). Starting from position 70,120,258 bp up to the end of the gene, the sequence of R182 perfectly aligned with the sequence of *S. lycopersicum*. Indeed, six SNPs and three INDELs were found between sequences from R182 and the *S. pennellii* genome in this last region (Figure 3).

## 4. Discussion

Since the last three decades, *S. pennellii* ILs were studied for identifying QTLs controlling different traits [5,8,47,48] including final yield, disease resistance and tolerance to abiotic stresses [7,49,50]. The ILs were obtained in the year 1992 through a series of backcrosses by the group of Eshed and Zamir [51,52], and then were again characterized by Fulton et al. (Tomato-EXPEN 2000) [53] and Chitwood et al. [6]. However, all these studies relied only on the use of the tomato genome of *S. lycopersicum* cv. Heinz as unique reference, though the IL parental lines derive from two species: *S. lycopersicum* and *S. pennellii*. This simplification could theoretically produce erroneous results when trying to precisely identify borders and extension size of the introgressed regions, considering that *S. pennellii* is a wild species with a genome quite different from *S. lycopersicum* in terms of size, collinearity, transposable elements, and so on [4]. For these reasons, a precise characterization of these introgression lines should take into consideration the two parental genomes to identify both the wild regions inserted (from the *S. pennellii* donor) and the cultivated region replaced (from the *S. lycopersicum* background).

Previously, the *S. pennellii* IL7-3 sub-line R182 was selected in our laboratory considering its good performances in terms of fruit quality [35]. This prompted us to better define genes mapping in R182 wild region, since this sub-line could be an important source of candidate genes and regulators controlling AsA synthesis and/or accumulation in red ripe fruit. In our previous study, we attempted to define the R182 introgressed region by designing 18 species-specific molecular markers using *S. lycopersicum* as reference genome [35].

The integration of genomics tools carried out here, as well as the comparison between the two parental genomes (*S. pennellii* and *S. lycopersicum*), allowed reaching two main aims of our work: (1) to exclude the presence of additional wild regions in the sub-line, which could be involved in AsA accumulation in the fruit, and (2) to exactly reconstruct the number and position of wild alleles that map in the introgressed region, following recombination events that occurred between the two parental genomes. First of all, both the genotyping procedures adopted in the present study (GBS and RNA-Seq variants detection) were conducted at genome-wide scale and allowed excluding that spurious wild fragments in the R182 sub-line could contribute to AsA synthesis and/or accumulation (Appendix A). Therefore, these results led us to only focus on the small-sized wild region present in the sub-line. Afterwards, we succeeded to finely define both the break points and the extension of the wild introgressed segment by investigating the variants detected from the RNA-Seq reads. This analysis allowed relying on a wide number of SNP-markers (>500 SNP in the putative introgressed region) that helped to precisely identify the interval of the wild genome in the sub-line. Additional molecular markers were also built on critical spots and the complete sequencing of a border-gene was also performed to accurately define the point where the introgression breaks.

The genome-based approaches performed helped to highlight that the introgressed region has a different size and a different number of annotated genes compared to both the parental lines *S. pennellii* and *S. lycopersicum*. In particular, we found that a 394 Kbp fragment including 33 *S. lycopersicum* genes (spanning from Solyc07g047990 to Solyc07g049310) was replaced by two segments of 284 Kbp and 164 Kbp including 23 (from Sopen07g024420 to Sopen07g024640) and 16 (from Sopen07g025130 to Sopen07g025280) *S. pennellii* genes, respectively. A synteny analysis between the parental lines demonstrated a translocation of the second wild segment (including 16 genes), which was embedded in the first region with 23 wild genes (Figure 2 and Appendix A). Although the hypothesis of a double recombination event is also possible, the short distance between the two wild segments make it more likely the hypothesis that a mis-assembly of a scaffold in that region of the *S. pennellii* genome occurred. Further investigation on the genome of *S. pennellii* will be necessary to ascertain this issue.

Since a recent update of the *S. lycopersicum* genome assembly and annotation was released [54], a more detailed study to shed light on the exact order of genes in the introgressed region of R182 was undertaken. The most recent release (iTAG4.1) just exhibits slight differences in the analyzed region when compared to the previous version v3.2. The main changes concern the genes Solyc07g048035, Solyc07g048085, and Solyc07g049165, which were removed from the iTAG4.1 annotation, the gene Solyc07g049135 that substituted the previous gene Solyc07g049140, and the gene Solyc07g049215, which was ex-novo annotated in the last release.

Among all the genes mapping in the introgressed region of R182, we focused our attention on the pairs Solyc07g049280/Sopen07g024610, Solyc07g049290/Sopen07g024620, and Solyc07g049310/Sopen07g024640. These genes, coding for a pyrophosphate-fructose 6-phosphate 1-phosphotransferase (PFP) and two major facilitator superfamily (MFS) membrane proteins, were already proposed as candidate genes [35] that could play an indirect role in AsA biosynthesis and accumulation, modulating the amount of AsA precursors. Indeed, the PFP is a key enzyme involved in the glycolysis/gluconeogenesis pathways [55], whereas MFSs perform the import or export of target substrates such as sugars [56]. The importance of MFSs was further confirmed by the results of the transcriptomics analysis performed on fruits at breaker and mature stages from the sub-line R182 and the cultivated genotype M82. The RNA-Seq analyses pointed out that both MFS genes were upregulated in R182. Therefore, these two genes might be good candidates for controlling AsA content in tomato fruit. Noteworthy, Sopen07g024640 and its ortholog Solyc07g049310, map on the border of the introgression. Our fine investigation of the lower border pointed out that the recombination event leading to the formation of the R182 sub-line occurred in the 3′UTR of this gene. This event produced a chimeric sequence of the gene (Figure 3), which combined most of the gene features of the *S. pennellii* allele (5′UTR, coding sequence and part of 3′UTR) with the remaining part of the 3′UTR of the *S. lycopersicum* form. This change/recombination of the 3′UTR could be responsible for the altered expression level of the gene Solyc07g049310, being UTRs important regulator regions for transcription [57]. The function and mechanisms of action of this gene will be validated by using genome editing techniques, such as the CRISPR/Cas9 technology.

Additional interesting findings were highlighted when comparing *S. lycopersicum* and *S. pennellii* genes belonging to the introgressed region of R182 (Figure 2). The integration of synteny and orthology analyses demonstrated that three cultivated genes, Solyc07g048020 annotated as charged multivesicular body, Solyc07g049135 annotated as fruit-specific protein, and Solyc07g049215 described as an unknown protein, do not seem to have any ortholog in the corresponding *S. pennellii* genome annotation. The absence of the Solyc07g049135 ortholog (also reported as Metallocarboxypeptidase inhibitor TCMP-2) into the genome of the R182 sub-line was also corroborated by the RNA-Seq analysis that showed a severe downregulation (almost no expression) of this gene in R182 (-13 logFC) at both fruit ripening stages (BR and MR) when compared to the M82 control line. In plant, the Metallocarboxypeptidase inhibitors, a type of proteases inhibitors (PI), are multifunctional proteins primarily involved in plant protection against pathogens and in plant tolerance modulation to diverse abiotic stresses. The production of PIs is highly regulated by different hormones, such as jasmonic acid, abscisic acid, and ethylene [58], which are reported to also regulate in turn ascorbic acid content [59,60,61,62,63]. In addition, a recent study reported that two specific metallocarboxypeptidase inhibitors in tomato, TCMP-1 and TCMP-2, were associated with fruit development [64]. The lack of this specific protease inhibitor in R182 line could, to some extent, influence the regulation of pathways that are linked to fruit development and ripening, maybe acting on the modulation of the hormonal network and, ultimately, on the production of AsA in the fruit.

Looking at the R182 wild fragment, we found seven *S. pennellii* specific genes (without any *S. lycopersicum* orthology), including a RNA-directed DNA polymerase protein (Sopen07g025250), a Shaker family potassium ion (K+) channel (Sopen07g025150), two cysteine-rich receptor-like protein kinases (Sopen07g024580 and Sopen07g025180) and three hypothetical proteins (Sopen07g024570, Sopen07g025190 and Sopen07g025200). Receptor-like kinases (RLKs) are ubiquitous molecular components and play essential roles in signal transduction by recognizing extracellular stimuli and activating downstream signaling pathways. A role for RLK was reported in environment responses [65], differentiation [66], growth, and development [67]. Even if no direct evidence of their involvement is reported for AsA accumulation, most CRKs genes are differentially expressed when the levels of reactive oxygen species (ROS) rise [68]. This highlights a possible role of RLKs in the oxido-reduction balance, which could probably imply an AsA homeostasis/modulation too.

Moreover, Sopen07g025260 and Sopen07g025270 align both with Solyc07g049240, representing a probable gene duplication in *S. pennellii*. Indeed, both Sopen07g024560 and Sopen07g025270 encode for a cold-inducible cationic peroxidase. Peroxidase are genes involved in the recycling pathway of ascorbate [33], which might have a role in shifting the final ratio of the oxidized/reduced AsA in tomato fruit.

Finally, since a consistent group of genes mapping outside the introgressed regions was differentially expressed, a more in-depth investigation of the introgressed region was performed in order to verify if some genes/TFs mapping within it could control the expression of genes in other portions of the genome. Among the genes mapping outside of the introgressed region of the sub-line R182, two genes Solyc03g097580 and Solyc05g024260, both belonging to the Bidirectional sugar transporter SWEET family and the Semi-SWEET-SWEET glucoside transporter superfamily, were upregulated in the breaker and mature stages, respectively, and the Solyc03g098290 encoding for a sucrose synthase was downregulated in the MR stage. All these genes could be involved in the AsA biosynthetic pathways and will be further investigated.

## 5. Conclusions

The genomic tools exploited to in depth reconstruct the introgressed region in the *S. pennellii* sub-line R182 allowed defining that 39 wild genes replaced 33 cultivated genes. In particular, a group of genes mapping within and out the introgressed region were identified as candidate genes controlling ascorbic acid content in the fruit of the sub-line, and are indirectly or directly involved in sugar and/or hormones pathways. Other wild genes encoding proteins with still unknown function could act as regulators of genes mapping outside the region and their potential role in this regulatory mechanism will be explored in the future. In addition, the chimeric structure of a Major facilitator superfamily (MFS) membrane protein mapping at the lower border of the introgression region deserves further investigation.

## Figures and Tables

**Figure 1 genes-11-00847-f001:**
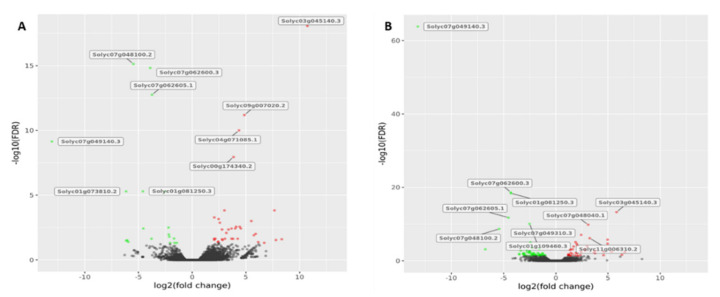
Volcano plots of genes differentially expressed between R182 and M82 at (**A**) breaker and (**B**) mature red fruit stages. The y-axis corresponds to the mean expression value of log 10 (false discovery rate, FDR) and the x-axis displays the log2 (fold change) value. The red dots represent the significantly upregulated genes while the green dots represent the significantly downregulated genes. Grey dots represent the genes whose expression levels did not reach statistical significance.

**Figure 2 genes-11-00847-f002:**
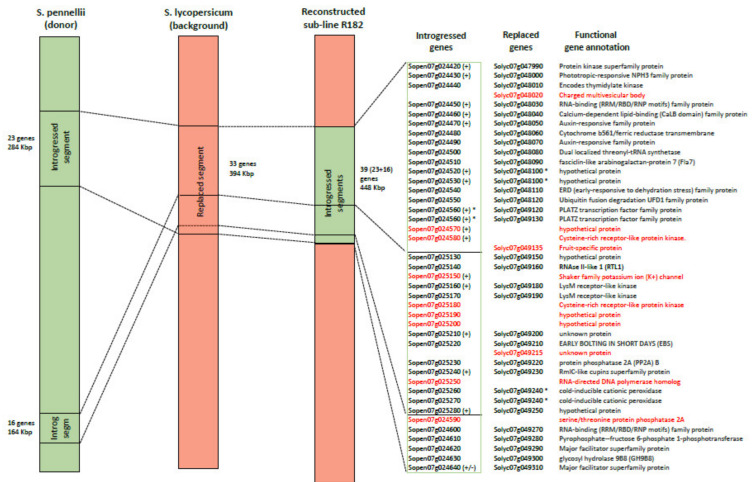
Reconstruction of the introgressed wild region in R182 by the synteny analysis of the genomic portion of chromosome 7 between *S. pennellii* and *S. lycopersicum*. The introgressed genes of *S. pennellii*, the replaced genes of *S. lycopersicum,* and their function annotation were highlighted in the right side of the figure. Genes evidenced in red represent species-specific genes. Experimentally tested markers (+) and duplicated genes (*) were reported next to the corresponding genes.

**Figure 3 genes-11-00847-f003:**
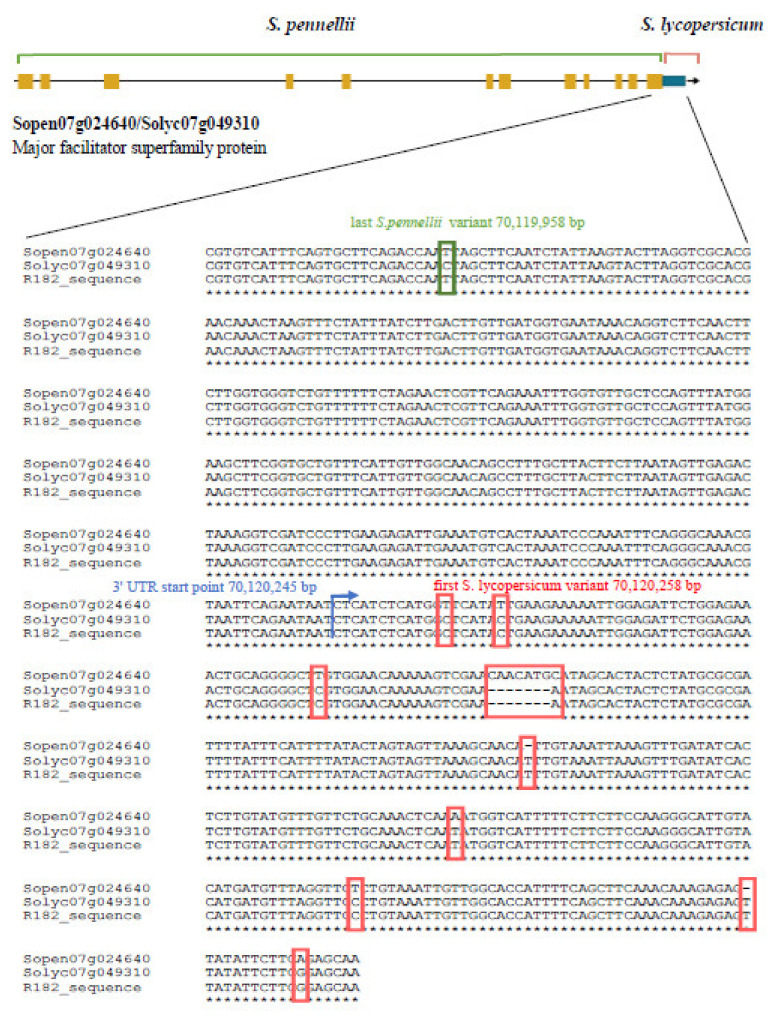
Multi-alignment of the last exon of Sopen07g024640, the ortholog Solyc07g049310 and the recombined sequence in R182. The recombination event occurred between the green rectangle showing the last *S. pennellii* variant of the R182 gene and the red rectangles showing the *S. lycopersicum* variants of R182 gene. The blue arrow points out the position where the 3’ Untranslated Region (UTR) starts.

**Table 1 genes-11-00847-t001:** Number of upregulated and downregulated genes between R182 and M82 at breaker (BR) and mature red (MR) stages. The distribution of the differentially expressed genes across the tomato chromosomes (Chr) is reported.

	R182 vs. M82 BR		R182 vs. M82 MR	
Chr	Upregulated	Downregulated	Upregulated	Downregulated
0	1	0	0	0
1	3	6	4	8
2	3	2	4	2
3	6	0	6	9
4	1	0	3	4
5	2	1	1	5
6	1	1	2	2
7	3	6	4	12
8	1	0	4	4
9	7	1	4	4
10	4	0	2	1
11	1	1	5	2
12	2	1	1	7
**Total**	35	19	40	60

**Table 2 genes-11-00847-t002:** Differentially expressed genes between R182 and M82 commonly identified at breaker (BR) and mature red (MR) fruit stages.

Gene ID ^1^	BR (LogFC)	MR (LogFC)	Position (SL3.0)	Functional Annotation
Solyc01g073810	−6.154	−2.521	81055373..81056169	LOW QUALITY: Cysteine/Histidine-rich C1 domain family protein
Solyc01g081250	−4.587	−4.299	80365913..80367306	Glutathione s-transferase, putative
Solyc03g045140	10.729	5.813	11633346..11652595	Cyclopropane-fatty-acyl-phospholipid synthase
Solyc03g096130	7.654	5.344	59424648..59428566	Protein yippee-like
Solyc03g096250	4.241	4.959	59662319..59665587	Protein yippee-like
Solyc05g021163	−3.790	−3.507	26616597..26617138	Ubiquitin-conjugating enzyme 34
Solyc07g048040	2.461	3.096	59358115..59359332	Calcium-dependent lipid-binding domain-containing protein
Solyc07g048100	−5.474	−5.426	59411341..59428966	BRCT domain-containing protein
Solyc07g049140	−13.070	−13.219	59495183..59496455	Metallocarboxypeptidase inhibitor
Solyc07g049200	−2.536	−1.398	59621580..59624273	Coiled-coil domain-containing protein 21, putative isoform 2
Solyc07g049310	3.003	2.426	59686834..59695592	Major facilitator superfamily protein
Solyc07g062600	−3.908	−4.307	65419623..65420348	Acyl-CoA N-acyltransferase with RING/FYVE/PHD-type zinc finger protein
Solyc07g062605	−3.745	−4.538	65423519..65426708	Acyl-CoA N-acyltransferase with RING/FYVE/PHD-type zinc finger protein
Solyc09g074380	2.161	1.885	66596363..66601065	DCD (Development and Cell Death) domain protein
Solyc11g011570	5.913	4.566	4638263..4639371	RING/U-box superfamily protein

^1^ Genes marked in bold belong to the introgression region of R182.

**Table 3 genes-11-00847-t003:** Homologous relationships between genes of *S. pennellii* (*S. pen*) (release v2) and *S. lycopersicum* M82 (*S. lyc*) (release iTAG 4.1) belonging to the introgressed region of R182 sub-line. For each gene, the position and the relationship type were reported.

*S. pennellii* Genes	Position (Start-End)	*S. lycopersicum* Genes	Position (Start–End)	Relationship Type
Sopen07g024420	69836210-69837672	Solyc07g047990	59085047–59086521	BBH
Sopen07g024430	69841308-69844381	Solyc07g048000	59090799–59093868	BBH
Sopen07g024440	69850538-69845963	Solyc07g048010	59095485–59099806	BBH
		Solyc07g048020	59102011–59102233	*S. lyc* specific
Sopen07g024450	69857939-69861502	Solyc07g048030	59104838–59109134	Homology
Sopen07g024460	69872065-69870851	Solyc07g048040	59117929–59119147	BBH
Sopen07g024470	69877402-69876191	Solyc07g048050	59122683–59123895	BBH
Sopen07g024480	69881212-69881726	Solyc07g048060	59127565–59128135	BBH
Sopen07g024490	69907539-69909885	Solyc07g048070	59139843–59142138	BBH
Sopen07g024500	69913450-69920824	Solyc07g048080	59154590–59161915	BBH
Sopen07g024510	69913450-69920824	Solyc07g048090	59163902–59164601	BBH
Sopen07g024520	69931348-69933371	Solyc07g048100	59171269–59183106	Probably duplicated in *S. pen*
Sopen07g024530	69933372-69943849	Solyc07g048100	59171269–59183106	Probably duplicated in *S. pen*
Sopen07g024540	69954645-69945131	Solyc07g048110	59184257–59193776	BBH
Sopen07g024550	69963499-69970741	Solyc07g048120	59201061–59208384	BBH
Sopen07g024560	69963499-69970741	Solyc07g049120	59262651–59264650	Probably duplicated in *S. lyc*
Sopen07g024560	69985256-69983275	Solyc07g049130	59271800–59273866	Probably duplicated in *S. lyc*
Sopen07g024570	70060300-70061901			*S. pen* specific
Sopen07g024580	70063119-70066285			*S. pen* specific
		Solyc07g049135	59285823–59287144	*S. lyc* specific
Sopen07g025130	70795423-70792524	Solyc07g049150	59334801–59337935	Homology
Sopen07g025140	70801910-70802955	Solyc07g049160	59338338–59343917	Homology
Sopen07g025150	70806557-70803357			*S. pen* specific
Sopen07g025160	70829776-70818613	Solyc07g049180	59357724–59365031	Homology
Sopen07g025170	70830302-70828932	Solyc07g049190	59367367–59368442	BBH
Sopen07g025180	70837317-70844338			*S. pen* specific gene
Sopen07g025190	70849697-70850207			*S. pen* specific gene
Sopen07g025200	70863988-70860217			*S. pen* specific gene
Sopen07g025210	70872807-70875563	Solyc07g049200	59414636–59417346	BBH
Sopen07g025220	70887686-70877146	Solyc07g049210	59417423–59423173	BBH
		Solyc07g049215	59423182–59423459	*S. lyc* specific
Sopen07g025230	70893995-70900317	Solyc07g049220	59429382–59435129	BBH
Sopen07g025240	70901562-70901131	Solyc07g049230	59435955–59436387	BBH
Sopen07g025250	70910568-70905495			*S. pen* specific gene
Sopen07g025260	70913034-70928413	Solyc07g049240	59441214–59442531	Probably duplicated in *S. pen*
Sopen07g025270	70941837-70943160	Solyc07g049240	59441214–59442531	Probably duplicated in *S. pen*
Sopen07g025280	70947923-70947081	Solyc07g049250	59443278–59444166	BBH
Sopen07g024590	70071789-70067951			BBH
Sopen07g024600	70075960-70073928	Solyc07g049270	59453256–59455454	BBH
Sopen07g024610	70084047-70076418	Solyc07g049280	59455644–59463344	BBH
Sopen07g024620	70087391-70084477	Solyc07g049290	59463688–59466693	BBH
Sopen07g024630	70094146-70090740	Solyc07g049300	59469239–59473026	BBH
Sopen07g024640	70111740-70120667	Solyc07g049310	59479929–59488648	Homology

BBH = Best Bidirectional Hits; Homology was computed by BLAST when no a direct BBH was found.

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
