# Peer review of "Genomic Dissection of a Wild Region in a Superior Solanum pennellii Introgression Sub-Line with High Ascorbic Acid Accumulation in Tomato Fruit"

_genes, 2020, doi:10.3390/genes11080847_

Round 1

Reviewer 1 Report

The research article by Aliberti et al. details the S. pennellii genomic introgression into M82 background using introgression line R182, which is a subline of IL7-3 with potentially high levels of ascorbic acid. A Genotyping-by-Sequencing approach accurately defined the recombination events on chromosome 7 and elsewhere in line R182, containing S. pennellii segments on a M82 S. lycopersicum background. The RNA-Seq approach highlighted putative important genes for ascorbic acid biosynthesis. Several of the most differentially expressed genes in this study fell within the introgression segment, which highlights their importance. Gene duplication events and non-orthologous genes were also reported as candidate key players in the introgressed region. The findings are interesting for future research that will identify the contribution of individual or groups of genes in ascorbic acid production and can be used in genome predictive projects in industry or academia.

The script lacks cohesion in writing style mainly at introduction and discussion. It is difficult to read at some parts. There are numerous unnecessary words (like “the”) that can be omitted or rewritten in a more direct way. For example in line 43-44: “All together, they represent the whole genome of the wild species, and are very useful to detect wild favourable alleles to be transferred in tomato cultivated genotypes”, could be rephrased as “An entire set of ILs represents a wild species’ entire genome and can be used to transfer traits into commercial cultivars.” Results and Methods and Materials are clearly written on the other hand.

Minor comments:

  • In line 49 “qualitative traits” can be replaced with “commercial traits of interest” as there is previous mentioning of QTLs in line 48 and someone can get confused with quantitative/qualitative (dominant) traits and not traits regarding fruit quality.
  • In line 101-103 the authors should mention what measures were taken in the open field to minimize cross-pollination by bees or other insects that might visit tomato flowers.
  • In line 106 and 125 there must be a clarification about the timing of harvest for leaves and fruits. If leaves were harvested before the fruits, then it will trigger a transcriptome change related to injury-responsive genes. The same should be detailed about the fruit harvesting on line 125. If fruits were harvested at 45 days post anthesis (dpa) and then again at 55 dpa from the same plant, then the first harvesting might affect the transcriptional profiling of the plant at 55 dpa compared to an unharvested plant at 55 dpa.
  • In line 169 “my Taq” is actually “MyTaqTM“.
  • In lines 177-179 the sequences of primers should go to a supplementary file.
  • In line 253 please delete the “33”.
  • In line 346 correct “critic” with “critical”.
  • Replace the word “couples” with “pairs” when referring to genes.

Author Response

Following you can find a point-by-point response to the reviewers' comments.

Reviewer 1:

The research article by Aliberti et al. details the S. pennellii genomic introgression into M82 background using introgression line R182, which is a subline of IL7-3 with potentially high levels of ascorbic acid. A Genotyping-by-Sequencing approach accurately defined the recombination events on chromosome 7 and elsewhere in line R182, containing S. pennellii segments on a M82 S. lycopersicum background. The RNA-Seq approach highlighted putative important genes for ascorbic acid biosynthesis. Several of the most differentially expressed genes in this study fell within the introgression segment, which highlights their importance. Gene duplication events and non-orthologous genes were also reported as candidate key players in the introgressed region. The findings are interesting for future research that will identify the contribution of individual or groups of genes in ascorbic acid production and can be used in genome predictive projects in industry or academia. The script lacks cohesion in writing style mainly at introduction and discussion. It is difficult to read at some parts. There are numerous unnecessary words (like “the”) that can be omitted or rewritten in a more direct way. For example in line 43-44: “All together, they represent the whole genome of the wild species, and are very useful to detect wild favourable alleles to be transferred in tomato cultivated genotypes”, could be rephrased as “An entire set of ILs represents a wild species’ entire genome and can be used to transfer traits into commercial cultivars.” Results and Methods and Materials are clearly written on the other hand.

Reply:We very much appreciate the comments, which were very constructive. We have revised the manuscript with improved language and proofreading through a native English speaking colleague. A specific effort was made in order to improve the cohesion in writing style along the entire manuscripts, paying much attention to introduction and discussion sections. All changes have been highlighted in yellow color and tracked.

Minor comments:

  • In line 49 “qualitative traits” can be replaced with “commercial traits of interest” as there is previous mentioning of QTLs in line 48 and someone can get confused with quantitative/qualitative (dominant) traits and not traits regarding fruit quality.

Reply: line 51, “qualitative traits” was replaced by “commercial traits of interest”.

  • In line 101-103 the authors should mention what measures were taken in the open field to minimize cross-pollination by bees or other insects that might visit tomato flowers.

Reply: Since tomato is considered a strict autogamous species with natural self-pollination varying from 94% to 99%, any specific measure was adopted to avoid an unlikely event of cross-pollination. In addition, molecular and phenotypic analyses of each genotype are in line with those expected, corroborating the purity of the lines.

  • In line 106 and 125 there must be a clarification about the timing of harvest for leaves and fruits. If leaves were harvested before the fruits, then it will trigger a transcriptome change related to injury-responsive genes. The same should be detailed about the fruit harvesting on line 125. If fruits were harvested at 45 days post anthesis (dpa) and then again at 55 dpa from the same plant, then the first harvesting might affect the transcriptional profiling of the plant at 55 dpa compared to an unharvested plant at 55 dpa.

Reply: Thank you for your comment. We used different plants of the same genotype to collect leaves and fruits. So, no injury response due to the harvesting of leaves was triggered in the transcriptome studied. Similarly, for the fruit harvesting, in order to avoid changes in the transcriptomic profile we harvested fruit at 45 and 55 dpa from different plants of the same genotype. We clarified this in the text, lines 118-122.

  • In line 169 “my Taq” is actually “MyTaqTM“.

Reply : line 185, we replaced my Taq with MyTaqTM

  • In lines 177-179 the sequences of primers should go to a supplementary file.

Reply: Thank you for your comments. We moved the primer sequences in a Supplementary Table (Table S1).

  • In line 253 please delete the “33”.

Reply: line 288, We have corrected the mistake

  • In line 346 correct “critic” with “critical”.

Reply : line 386, We have corrected “critic” with “critical”

  • Replace the word “couples” with “pairs” when referring to genes.

Reply : line 409, We have corrected “couples” with “pairs”

Reviewer 2 Report

The paper by Aliberti et al presents data on the characterisation of an introgression line (S.pennellii in M82) which has improved agronomic characteristics, including for vitamin C content.

The statement line 61/62 is not entirely true. The ascorbic acid synthesis pathway has been well characterised and genes and mutants identified. Although probably not all upstream regulators are known.

For general information and introduction it would be good to include a summary of the phenotypes of the R182 line compared to M82 and/or IL7.3.

Figure 1 is hard to follow and needs to be improved, 1B needs a key for the two colours. I am not sure about the point Solyc07g049140 which appears to be an extreme outlier : have the raw data been checked ? More information should be included in legend to aid the reader in understanding – for example « up and down regulated genes » means up and down regulated in what compared to what ?

More generally I did not understand why the study had been carried out on MR and BR fruit tissue. What happened to the leaf samples described in the materials and methods ? Leaf expressed genes and also affect fruit phenotypes and further candidate genes could be found.

I also did not understand why there was so much focus on differentially expressed genes. It was mentioned at the beginning of the results that 4 genes of the region contained SNPs or INDEL that affected protein sequence. Why were these not considered more in the study as potential candidates ? Or indeed any other SNP arising from the RNAseq analysis.

Author Response

Following you can find a point-by-point response to the comments

  • The paper by Aliberti et al presents data on the characterisation of an introgression line (S.pennellii in M82) which has improved agronomic characteristics, including for vitamin C content. The statement line 61/62 is not entirely true. The ascorbic acid synthesis pathway has been well characterised and genes and mutants identified. Although probably not all upstream regulators are known.

Reply : Thank you for your suggestions. We modified the statement (lines 67-69) as follow: “As for AsA, the biosynthetic pathways controlling its synthesis and accumulation are well characterized, but not all upstream regulators in tomato are known.”

  • For general information and introduction it would be good to include a summary of the phenotypes of the R182 line compared to M82 and/or IL7.3.

Reply : We appreciate this comment, which helped improving our manuscript. We modified the text (lines 92-95) to include a short summary of the main phenotype of R182 compared to M82 and IL7-3.

  • Figure 1 is hard to follow and needs to be improved, 1B needs a key for the two colours. I am not sure about the point Solyc07g049140 which appears to be an extreme outlier : have the raw data been checked ? More information should be included in legend to aid the reader in understanding – for example « up and down regulated genes » means up and down regulated in what compared to what ?

Reply : Thank you for your suggestions. In order to improve the readability of the figure we decided to transfer data reported in Figure 1A and 1B in a table (Table 1) and leave in the figure only data reported in  1C and 1D. We also improved the caption to allow a better understanding of the figure.

About the Solyc07g049140, we carefully checked the result and raw data and the gene resulted highly down-regulated in the all the three replicates. We can confirm the reliability of the data presented (see Supplementary Table 4 for all the details related to the expression of this gene).

  • More generally I did not understand why the study had been carried out on MR and BR fruit tissue. What happened to the leaf samples described in the materials and methods ? Leaf expressed genes and also affect fruit phenotypes and further candidate genes could be found.

Reply : Thank you for the comment. Since the main phenotype observed in R182 (increased ascorbic acid, high sugar content, etc) were assessed for fruit at two specific stages (breaker and mature red), we performed RNA-Seq analysis on the same stages as well. By contrast, leaf tissue was used just to obtain DNA for performing the GBS analysis. We agree with the reviewer that studying the mechanisms affecting metabolites translocation from leaf to fruit may be an interesting point. However, according to our knowledge and as reported by other authors (Badejo et al. 2012) AsA translocation from leaf to fruit was an active process in the first developmental stages of fruit while its contribution after breaker stage (as in our case) was not relevant.

  • I also did not understand why there was so much focus on differentially expressed genes. It was mentioned at the beginning of the results that 4 genes of the region contained SNPs or INDEL that affected protein sequence. Why were these not considered more in the study as potential candidates ? Or indeed any other SNP arising from the RNAseq analysis.

Reply : Thank you for the comment. The results of this work aimed both at identifying the main genomic differences between the wild and cultivated regions underlying the introgression of R182 and at characterizing the gene expression changes in comparison with the M82 parental line. For the first point, we observed some major genomic changes, like duplication or deletion events, structural rearrangement of one gene as well as changes in the number of genes between wild and cultivated genomes. For this reason, we decided to focus on these major changes more than on specific SNP/INDEL. However, according to the four genes of the region containing SNPs or INDEL, the SNPeff analysis (which predict the impact of the variant on the protein functionality) highlighted that almost all the variants have a “MODIFIER” impact (that means not possible to determine) since they fall into intron or intergenic regions. Only one variant was reported to have a “LOW” impact and it was annotated as a synonymous change (no impact on the protein). However, all the details about SNP and INDEL of the introgressed region identified through the GBS approach are reported in Supplementary Table 1. 

Round 2

Reviewer 2 Report

The authors have responded to most of my comments. For the differences between the lines R182 compared to M82 and IL7-3 (modification lines 92-95), I hoped to see quantifiable data for example R182 contains x% more ascorbate, has higher yield by y% etc.

Author Response

Following a point-by-point response to the reviewer's comment.

1) The authors have responded to most of my comments. For the differences between the lines R182 compared to M82 and IL7-3 (modification lines 92-95), I hoped to see quantifiable data for example R182 contains x% more ascorbate, has higher yield by y% etc

Reply : We thank the reviewer for the constructive remarks. We added a sentence to better define the R182 phenotype compared to the parental lines (lines 147-149 of the revised manuscript). We also checked the manuscript to further improve style and to correct minor spell errors (highlighted in green).